# Metal-organic framework glasses with permanent accessible porosity

Chao Zhou [1,2], Louis Longley[1], Andraž Krajnc[3], Glen J. Smales[4,5], Ang Qiao [6], Ilknur Erucar[7], Cara M. Doherty[8], Aaron W. Thornton[8], Anita J. Hill[8], Christopher W. Ashling [1], Omid T. Qazvini[9], Seok J. Lee[9], Philip A. Chater [5], Nicholas J. Terrill [5], Andrew J. Smith [5], Yuanzheng Yue[2,6,10], Gregor Mali [3], David A. Keen [11], Shane G. Telfer [9] & Thomas D. Bennett [1]

To date, only several microporous, and even fewer nanoporous, glasses have been produced, always via post synthesis acid treatment of phase separated dense materials, e.g. Vycor glass. In contrast, high internal surface areas are readily achieved in crystalline materials, such as metal-organic frameworks (MOFs). It has recently been discovered that a new family of melt quenched glasses can be produced from MOFs, though they have thus far lacked the accessible and intrinsic porosity of their crystalline precursors. Here, we report the first glasses that are permanently and reversibly porous toward incoming gases, without post-synthetic treatment. We characterize the structure of these glasses using a range of experimental techniques, and demonstrate pores in the range of 4 – 8 Å. The discovery of MOF glasses with permanent accessible porosity reveals a new category of porous glass materials that are elevated beyond conventional inorganic and organic porous glasses by their diversity and tunability.

[1] Department of Materials Science and Metallurgy, University of Cambridge, Charles Babbage Road, Cambridge CB3 0FS, UK. [2] Department of Chemistry and Bioscience, Aalborg University, DK-9220 Aalborg, Denmark. [3] Department of Inorganic Chemistry and Technology, National Institute of Chemistry, SI-1001 Ljubljana, Slovenia. [4] Department of Chemistry, University College London, Gordon Street, London WC1H 0AJ, UK. [5] Diamond Light Source Ltd, Diamond House, Harwell Science and Innovation Campus, Didcot OX11 0DE, UK. [6] State Key Laboratory of Silicate Materials for Architectures, Wuhan University of Technology, Wuhan 430070, China. [7] Department of Natural and Mathematical Sciences, Faculty of Engineering, Ozyegin University, Istanbul, Turkey. [8] Future Industries, Commonwealth Scientific and Industrial Research Organisation, Clayton South, VIC 3168, Australia. [9] MacDiarmid Institute for Advanced Materials and Nanotechnology, Institute of Fundamental Sciences, Massey University, Palmerston North 4442, New Zealand. [10] School of Materials Science and Engineering, Qilu University of Technology, Jinan 250353, China. [11] ISIS Facility, Rutherford Appleton Laboratory Harwell Campus, Didcot, Oxon OX11 0QX, UK. Correspondence and requests for materials should be addressed to S.G.T. (email: S.Telfer@massey.ac.nz) or to T.D.B. (email: tdb35@cam.ac.uk)

Conventional porous glasses are silica-rich materials derived by acid treatment of phase-separated borosilicate glass precursors. They have found widespread applications in electrodes, chromatography and medical devices, and as desiccants, coatings and membranes[1]. In general, they are easily processed and exhibit high mechanical stability. However, these advantages are offset by limitations in their pore sizes, which are typically confined to the macroporous size regime. No porous glass has been fabricated directly by melt-quenching, without post-treatment. Few microporous glasses have been reported, although advances in their synthesis would be particularly attractive for applications based on the selective uptake of gases and small molecules[2]. One of the most well-studied microporous glasses is Vycor, which is obtained by first melt-quenching a borosilicate liquid, and then heat treating the glass at a temperature well above $T_g$ to encourage phase separation. The borate-rich phase is then leached out via acid treatment to generate pores of over 3 nm radii and Brunauer–Emmett–Teller (BET) surface areas in the range of 100–200 $m^2\,g^{-1}$ [1]. Silica-derived aerogels have also attracted much attention due to their own high BET surface areas and pore sizes of between 0.5 nm and 1 nm[3]. The desire to introduce controllable properties based on organic functional groups has however also spurred the development of amorphous microporous organic materials, such as polymer of intrinsic microporosity-1 (PIM-1)[4,5]. Surface areas verging on 1000 $m^2\,g^{-1}$ arising from pore sizes of up to 1 nm have been reported. While these organic materials present fewer chemical restrictions on their functionality than their inorganic counterparts, their thermal stability is unfortunately limited and their susceptibility to densification over time (physical aging) is well documented[5].

In this context, porous materials that combine the advantageous properties of both inorganic and organic glasses are highly sought after. Recent advances in our understanding of the flexible behaviour[6] of soft porous metal–organic frameworks (MOFs)[7] have combined with a surge in research into defects[8] and disorder[9] to redefine our perception of MOFs as perfect crystalline materials[10]. We, and others, have drawn on these advances to show that crystallinity is not a prerequisite for many of the attractive properties of MOFs, and to produce liquids and glasses with internal cavities and ion conducting abilities[11–13].

One sub-set of MOFs that has garnered much attention is the zeolitic imidazolate framework (ZIF) family, numbering over 140 distinct structures containing tetrahedral transition metal ions, linked by imidazolate bis(monodentate) ligands into zeolitic architectures[14,15]. Several members of the family melt upon heating to around 400 °C. For example, in the case of certain polymorphs of ZIF-4 ([Zn(Im)$_2$], Im = imidazolate, $C_3H_3N_2^-$), this process proceeds on a sub-nanosecond timescale near $T_m$ (melting temperature) and results in a viscous MOF liquid of identical chemical composition to the crystalline solid[11]. Quenching of the liquid results in a frozen structure containing tetrahedral Zn(II) ions bridged by Im ligands. The continuous random network produced is similar to that of $aSiO_2$, but with the advantage of having both inorganic and organic components[11,16].

Known hybrid glasses, formed from ZIFs and phosphate coordination polymers[17,18], are all inaccessible to guest molecules. Here, motivated by the prospect of combining the attractive properties of both inorganic and organic moieties in a porous glass, we sought to develop *accessible* porosity in a melt-quenched MOF glass. We herein report the successful realization of this goal, in the form of glasses derived from ZIFs that reversibly adsorb gas molecules. These materials can be considered prototypical new porous glasses, given that there is no requirement for post-processing treatment. They are synthesized using a straightforward protocol and are stable, retaining their porosity in air over extended periods. These results provide a rational strategy upon which other porous MOF liquid and glass systems can be produced. We anticipate that the stability, processability and chemical diversity of these glasses will underpin their applications in separations, and as components of membranes, catalysts, functional coatings and thin films.

## Results

**Thermal characterization and vitrification of ZIF precursors.** The free energy requirement for breaking the zinc-imidazolate bond has been calculated to be ca. 95 kJ mol$^{-1}$ at 567 °C[11]. This relatively low bond strength led us to consider ZIFs containing this ligand as primary candidates for melting, followed by quenching of the liquid to produce porous glasses. Limitations exist to the exclusive use of imidazolate however, since close association of the Zn and Im components in the liquid and glass states leads to dense materials[16]. We thus developed a strategy involving the incorporation of sterically congested benzimidazolate (bIm) ligands into the framework, alongside the parent imidazolate linker. We anticipated that the inclusion of a bulky ligand would prevent close packing of the liquid state, and therefore facilitate pore network formation upon quenching. Similar approaches have proven successful in optimizing porosity in both crystalline materials[19] and amorphous organic polymers[5]. Suitable levels of the imidazolate linker are however still required to facilitate melting, since frameworks constructed exclusively from benzimidazolate ligands do not melt.

To develop our strategy, we initially focused on ZIF-76, [Zn (Im)$_{1.62}$(5-ClbIm)$_{0.38}$]      (5-ClbIm = 5-chlorobenzimidazolate, $C_7H_4N_2Cl^-$)[20]. This three-dimensional framework, formed from Zn$^{2+}$ nodes connected by Im or 5-ClbIm ligands (Fig. 1a), possesses the zeolitic LTA (Linde Type A) topology. In this network architecture, sodalite cages are connected together by rings of 4 Zn ions linked to one another and give rise to a supercage in the centre of the cell[21]. The structure possesses a framework density of 1.03 tetrahedral atoms per nm$^3$. This is extremely low when compared with ZIF-4 (3.66), which forms a non-porous glass, and ZIF-8 [Zn(mIm)$_2$], (mIm = 2-methylimidazolate, $C_4H_5N_2^-$) (2.45), which does not melt[22]. The structure of ZIF-76 was confirmed by powder X-ray diffraction, and the Im/5-ClbIm ratio determined by $^1$H nuclear magnetic resonance (NMR) spectroscopy on dissolved samples. Desolvation of ZIF-76 resulted in loss of occluded 5-chlorobenzimidazole from the pores (Supplementary Figures 1–5).

Thermogravimetric analysis (TGA) performed on an evacuated sample of ZIF-76 under argon resulted in a featureless trace until thermal decomposition at around 515 °C. An accompanying differential scanning calorimetry (DSC) measurement, which quantifies the heat absorption of a sample relative to a reference, revealed an endothermic feature that we ascribe to a melting transition at 451 °C ($\Delta H_f = 1.8$ kJ mol$^{-1}$) (Fig. 1b). Cooling of this liquid to room temperature produced a vitreous product, termed a$_g$ZIF-76 (Supplementary Figure 1). This terminology identifies an amorphous (a) product formed by melting, then quenching (subscript g referring to a glass), crystalline ZIF-76. The a$_g$ZIF-76 possesses an identical chemical composition to ZIF-76 and (most likely) has the same short range chemical connectivity as that of the ZIF-76 crystal structure. The a$_g$ZIF-76 exhibits a glass transition temperature, $T_g$, of 310 °C upon reheating in a DSC measurement (Fig. 1b, Supplementary Figure 6).

The isostructural crystalline framework [Zn(Im)$_{1.33}$ (5-mbIm)$_{0.67}$] (5-mbIm = 5-methylbenzimidazolate, $C_8H_7N_2^-$), referred to as ZIF-76-mbIm, also forms a glass (a$_g$ZIF-76-mbIm) (Supplementary Figure 7)[23,24]. A comparison of these with

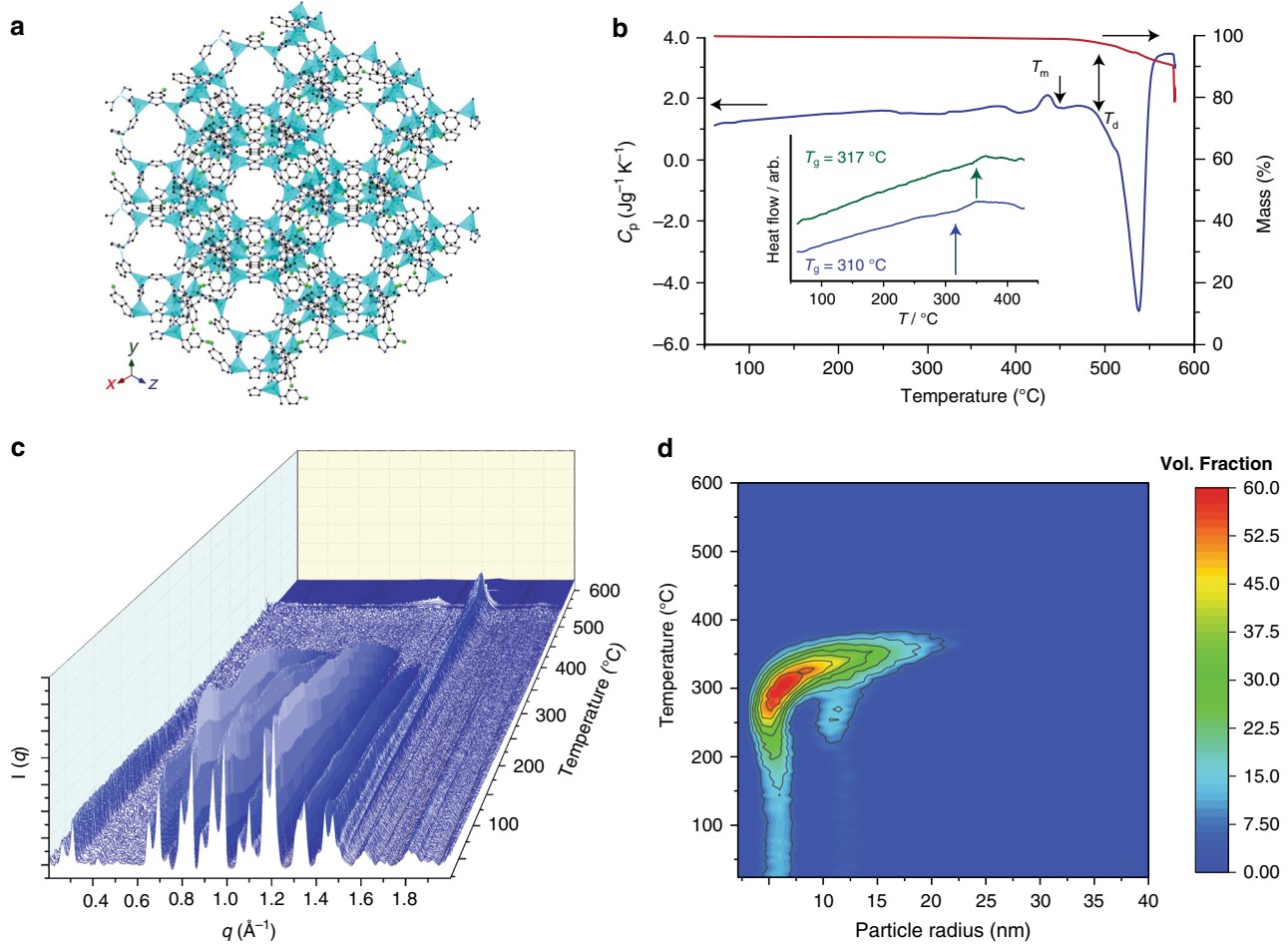

**Fig. 1** Liquid and glass formation of ZIF-76. **a** The structure of ZIF-76, as determined by single-crystal X-ray diffraction[20]. Zn light blue, Cl green, C grey, N dark blue, H omitted for clarity. **b** The isobaric heat capacity ($C_p$) and mass change (%) of ZIF-76 measured during a DSC-TGA upscan at 10 °C min$^{-1}$, highlighting the stable liquid domain between $T_m$ and $T_d$. Inset shows glass transitions for a$_g$ZIF-76 (blue) and a$_g$ZIF-76-mbIm (green). **c** Temperature resolved WAXS profile of ZIF-76 upon heating from 25 °C to 600 °C. Colour shading is included as a guide to the eye. **d** Temperature resolved volume fraction distributions of the different particle sizes of ZIF-76, indicating coalescence into particles of up to 30 nm

ZIF-76 and a$_g$ZIF-76 allowed us to study the relationship between ZIF structure and the properties of the resulting glass. ZIF-76-mbIm melts at 471 °C (Supplementary Figure 8), indicating a higher energy barrier to ligand dissociation due to the electron-donating methyl group. A similar electronic effect was previously suggested as a reason why ZIF-8 may possess a higher melting temperature than the framework decomposition temperature[11]. The a$_g$ZIF-76-mbIm possesses a $T_g$ of 317 °C (Fig. 1b, Supplementary Figure 9). The linker ratio of ZIF-76-mbIm, and indeed ZIF-76, is maintained in their respective glasses (Table 1, Supplementary Figures 2-5, 10 and 11), which is expected on the basis that no mass loss occurs during the melting process. The greater $T_g$ for a$_g$ZIF-76-mbIm is consistent with the greater van der Waals radius of a methyl group compared to chlorine atom, which leads to stronger non-covalent interactions between the framework constituents. This trend is reminiscent of the relationship between $T_g$ and steric side groups in organic polymers[25,26].

The transformation of ZIF-76 from the crystalline to the liquid state was investigated by small- and wide-angle X-ray scattering (SAXS and WAXS). Bragg diffraction in the variable temperature WAXS pattern decreases sharply around 420 °C, consistent with the melting point from DSC. The resultant diffuse scattering then ceases at ca. 520 °C, corresponding to thermal decomposition

(Fig. 1c). The liquid formed from ZIF-76 is therefore stable over a temperature range of approximately 100 °C, consistent with TGA evidence (Fig. 1b).

The SAXS intensity follows a power law behaviour of the form Q$^{-\alpha}$ at room temperature (Supplementary Figures 12 and 13). Fitting gave a value of $\alpha = 3.7$, which decreased to 3.5 at 360 °C, indicating a roughening of internal surface structure upon melting. Similar results were observed in a previous study of the melting of a different material, ZIF-62[27]. The dynamic nature of the liquid phase upon heating was evidenced by a steady increase in $\alpha$ to 3.9 between 360 °C and 500 °C, which we associate with a regularization of the internal pore structure of the liquid with both time and temperature. The discontinuity in the SAXS profile at around 550 °C indicates thermal decomposition, in accordance with the WAXS and DSC measurements. The volume weighted fractions of the particles below the observable limit of 315 nm in the sample were extracted from the data (Fig. 1d), showing crystalline ZIF-76 particles of 5 nm, 12 nm and 20 nm radius, which coalesce at ca. 370 °C before melting at higher temperatures.

**Linker positioning in the vitreous state**. The chemical environments, distribution and relative motion of the framework linkers in the liquid state prior to vitrification were studied to gain

**Table 1 Summary of the crystalline and glass samples**

|  | Composition | State | $T_m$ (°C) | $T_g$ (°C) | $T_d$ (°C) |
|---|---|---|---|---|---|
| ZIF-76 | $[Zn(Im)_{1.62}(5\text{-ClbIm})_{0.38}]$ | Crystalline | 451 | – | 517 |
| $a_g$ZIF-76 | $[Zn(Im)_{1.62}(5\text{-ClbIm})_{0.38}]$ | Glass | – | 310 | 511 |
| ZIF-76-mbIm | $[Zn(Im)_{1.33}(5\text{-mbIm})_{0.67}]$ | Crystalline | 471 | – | 596 |
| $a_g$ZIF-76-mbIm | $[Zn(Im)_{1.33}(5\text{-mbIm})_{0.67}]$ | Glass | – | 317 | 590 |

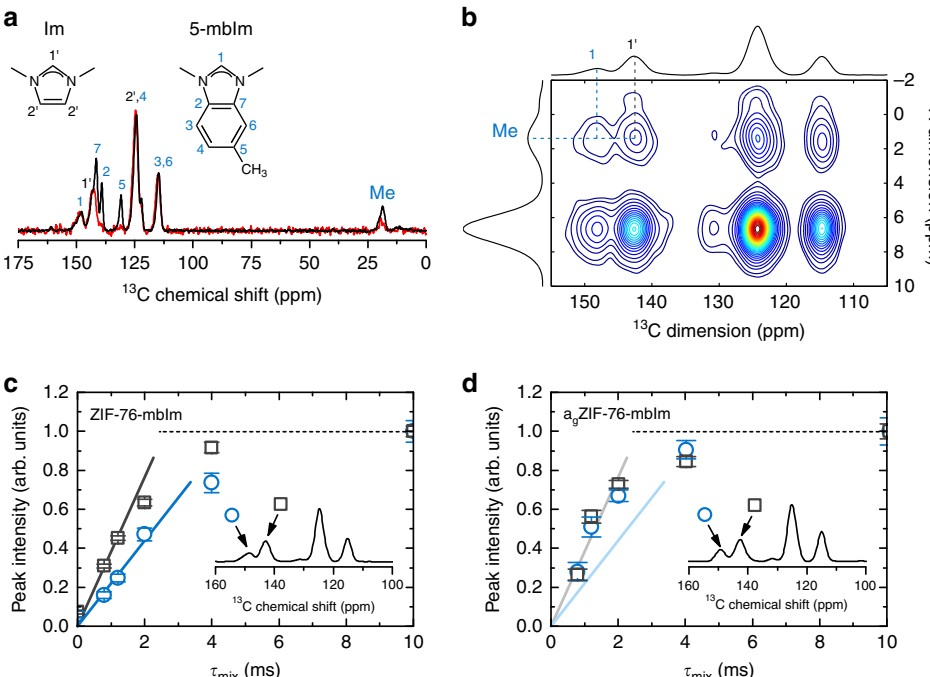

**Fig. 2** Linker dynamics upon melting. **a** The $^{13}$C MAS (black solid line) and $^1$H-$^{13}$C Lee–Goldburg cross-polarization MAS (red solid line) NMR spectra of crystalline ZIF-76-mbIm. Tentative assignment of individual signals is based on literature data for isotropic chemical shifts from the two molecular fragments. The assignment is further confirmed by comparing the resonances in the $^{13}$C MAS and $^1$H-$^{13}$C Lee–Goldburg cross-polarization MAS NMR spectra. In the latter, only resonances belonging to those carbons with hydrogens attached can be observed. **b** The $^{13}$C-detected 2D spin-diffusion NMR spectrum, recorded with a mixing time of 10 ms. Cross peaks due to polarization transfer between methyl protons of 5-mbIm on one hand and H1′ protons of Im and H1 protons of 5-mbIm on the other are denoted. **c**, **d** Spin-diffusion curves from the $^{13}$C-detected measurements on ZIF-76-mbIm and $a_g$ZIF-76-mbIm (squares: inter-linker polarization transfer between methyl protons of 5-mbIm and H1′ protons of Im; circles: intra-linker polarization transfer between methyl and H1 protons of 5-mbIm). Solid lines in **c** and **d** indicate the initial slopes of the two spin-diffusion curves for crystalline ZIF-76-mbIm (i.e., lines in **d** are equal to lines in **c**)

insight into phase separation processes and glass homogeneity. Solid-state $^1$H magic-angle spinning (MAS) NMR spectroscopic measurements were carried out for this purpose. Measurements on an evacuated sample of ZIF-76 yielded a single, broad, unresolvable signal at 6–7 ppm arising from both Im and ClbIm protons (Supplementary Figure 14). The spectrum of ZIF-76-mbIm featured an additional peak at 1.5−2 ppm, belonging to the methyl substituent of the benzimidazolate ring. The $^{13}$C MAS NMR spectra of these crystalline samples exhibited several partly resolved signals, matching those expected from the chemical structure of the ligand (Supplementary Figure 15). Vitrification resulted in little change to the resonances in both the $^1$H and $^{13}$C spectra, aside from a small shift in position of the methyl group resonance to lower field in the $^{13}$C NMR spectrum of ZIF-76-mbIm. These MAS NMR results demonstrate that the chemical environments of the ligands in the glasses are broadly similar to those in their crystalline precursors.

The distribution of the organic linkers within both crystal and glass phases was also investigated using spin-diffusion NMR

spectroscopy. This is a powerful technique which is able to detect the proximity of different organic components based on the rate of polarization transfer between their protons[28,29]. $^1$H-detected spin-diffusion experiments were first performed on ZIF-76-mbIm to track the transfer of proton polarization from the methyl group protons of 5-mbIm (resonance at 2 ppm) to all other protons in the sample (with chemical shifts in the region 6–7 ppm). Off-diagonal peaks at a mixing time of 0 ms are absent, as expected. Their appearance and gradual strengthening in intensity after this time, however, is due to polarization transfer (Supplementary Figure 16). After mixing times of 10–12 ms, a plateau in the spin-diffusion curve (Supplementary Figure 17) was observed, indicating that maximum transfer to both imidazolate and other 5-methylbenzimidazolate protons had been achieved. The detection of a well-resolved and isolated $^1$H signal from the methyl group protons of 5-mbIm and the $^{13}$C signal from **C1′** of the Im ligand (Fig. 2a) facilitated $^{13}$C-detected proton spin-diffusion measurements. This allowed us to track the polarization transfer from the methyl group to **H1** protons of the 5-

methylbenzimidazolate (intra-linker transfer) and to **H1'** protons of the imidazolate (inter-linker transfer between the two different linkers) (Fig. 2b). In crystalline ZIF-76-mbIm, the inter-linker transfer of polarization was observed to be faster than intra-linker transfer, implying that the linkers are well mixed within the framework (Fig. 2c). In $a_g$ZIF-76-mbIm, very fast spin diffusion among the linkers of two different types confirms that the 5-methylbenzimidazolate and imidazolate linkers remain very well mixed and do not separate into domains. Notably, the initial parts of the spin-diffusion curves of the glass are steeper than in the crystal, suggesting that the inter-linker distances between 5-methylbenzimidazolate and imidazolate linkers contract slightly upon vitrification (Fig. 2d). This is consistent with the downfield shift of the methyl group resonance in the $^{13}C$ NMR spectrum of $a_g$ZIF-76-mbIm, which may arise from closer contacts with the π electron clouds of neighbouring ligands.

**Pair distribution function measurements**. To further probe the structures of the ZIFs before and after vitrification, synchrotron X-ray total scattering measurements were performed on ZIF-76, ZIF-76-mbIm, $a_g$ZIF-76 and $a_g$ZIF-76-mbIm (Fig. 3a). The glassy nature of $a_g$ZIF-76 and $a_g$ZIF-76-mbIm was confirmed by the absence of sharp features in their respective structure factors. The corresponding pair distribution functions (PDFs), which are in effect atom–atom distance histograms, of all samples were extracted after appropriate data corrections (Fig. 3b). The PDFs of all samples are near identical for distances up to 6 Å. Since the correlations in this range are C-C/C-N (1.3 Å) and Zn-N (2 Å and 4 Å), this implies that the local $Zn^{2+}$ environment is near identical in all four samples. The limit of this order, at 6 Å, corresponds to the distance between two $Zn^{2+}$ centres, and confirms that metal–ligand–metal connectivity is present within both

crystalline and glass samples. This is consistent with observations made on other MOF glasses, and it underscores the consistency in the composition of these materials and the similar coordination environment of the zinc(II) ions across the crystalline and amorphous phases[16].

The PDFs of dense ZIF glasses are relatively featureless beyond 6 Å[16], which contrasts with the PDFs of $a_g$ZIF-76 and $a_g$ZIF-76-mbIm. In particular, the feature at 7.5 Å in the $D(r)$ for $a_g$ZIF-76 (Peak A, Fig. 3b and inset) (and for $a_g$ZIF-76-mbIm, Supplementary Figure 18) is related to a $Zn-N^3$ distance, where $N^3$ is the third nearest N atom to a given $Zn^{2+}$ ion. A further feature at 10.8 Å in the $D(r)$ for $a_g$ZIF-76 is related to the distance between a $Zn^{2+}$ ion and the Cl group on the next nearest neighbour ligand (Peak B, Fig. 3b and inset). The observation of medium range order (MRO) in the glass state, i.e., Zn–Im–Zn–ClbIm connectivity, is ascribed to the relative sluggish diffusion kinetics and high viscosity of the liquid phase. Lower viscosities associated with greater ligand movement would result in vastly reduced Zn–Im–Zn–ClbIm correlations[11]. The extended connectivity here would also render a contiguous pore network possible, as confirmed later by gas adsorption. Similar analysis of the PDFs for ZIF-76-mbIm and $a_g$ZIF-76-mbIm (Supplementary Figure 18) also indicates a degree of MRO in the glass, with oscillations persisting to higher $r$ values. This is also consistent with the lower position of the first sharp diffraction peak of ZIF-76-mbIm[30].

Synchrotron X-ray diffraction data were collected during the melting process of ZIF-76 (Fig. 3c). We observed a reduction in Bragg intensity upon heating, resulting in complete loss of all sharp diffraction peaks in the $S(Q)$ at the highest studied temperature of 540 °C. The amorphous pattern is recovered to room temperature on cooling; sharp Bragg peaks do not reappear. In the PDF, features at $r$ values of over 8–10 Å are present in

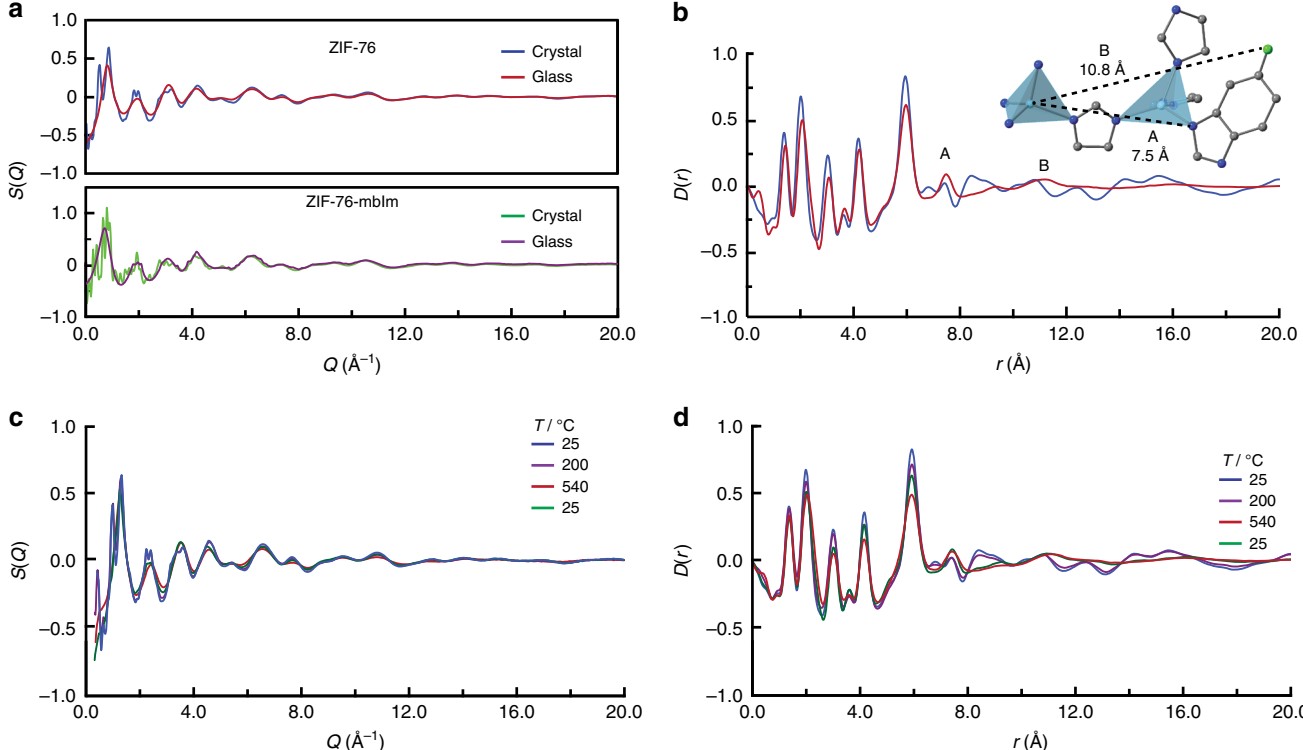

**Fig. 3** Diffraction in the crystalline and glass states. **a** X-ray structure factors $S(Q)$ of crystalline ZIF-76 and ZIF-76-mbIm, along with the corresponding glass samples. **b** Corresponding ZIF-76 pair distribution function $D(r)$. Inset shows medium range order. Zn light blue, Cl green, C grey, N dark blue, H omitted for clarity. **c, d** X-ray structure factors $S(Q)$ and pair distribution functions $D(r)$, respectively, of ZIF-76 upon heating and subsequent recovery to room temperature (green curve at 25 °C)

crystalline ZIF-76, but are virtually absent by 540 °C and in the sample at room temperature after heating (Fig. 3d).

**Accessible porosity in a MOF glass.** We have previously shown that positron annihilation lifetime spectroscopy (PALS) is a useful tool for the mapping of various pore sizes in MOF crystals and glasses[31]. In agreement with the crystal structure of ZIF-76[20], two cavity diameters of 5.7 Å and 15.7 Å were detected in our ZIF-76 samples (Supplementary Figure 19, Supplementary Table 1). Upon vitrification to $a_g$ZIF-76, a single pore with a diameter of ca. 5 Å was revealed. PALS analysis of crystalline ZIF-76-mbIm also yielded two cavities of diameters 5.8 Å and 15.6 Å. The $a_g$ZIF-76-mbIm glass retains two distinct pores with diameters of 4.8 Å and 7.2 Å (Supplementary Figure 20). Although this analysis shows a reduction in porosity due to vitrification, the glasses, and particularly $a_g$ZIF-76-mbIm, maintain significant porosity.

To prepare samples possessing high pore volumes before conducting gas adsorption experiments, we optimized the synthetic procedures for ZIF-76 and ZIF-76-mbIm, and incorporated larger quantities of the benzimidazolate-derived linker in each case. This strategy borrows the logic employed by Yaghi and colleagues[19] in the synthesis of new high surface area crystalline materials. Samples of ZIF-76 and ZIF-76-mbIm, with stoichiometries of $[Zn(Im)_{1.0}(5\text{-}ClbIm)_{1.0}]$ and $[Zn(Im)_{0.93}(5\text{-}mbIm)_{1.07}]$, respectively, were thus prepared. The ligand ratios in these materials were determined by $^1H$ NMR spectroscopy on dissolved samples (Supplementary Figures 21 and 22). A range of gas adsorption isotherms were measured on crystalline $[Zn(Im)_{1.0}(5\text{-}ClbIm)_{1.0}]$ and $[Zn(Im)_{0.93}(5\text{-}mbIm)_{1.07}]$. From $N_2$ adsorption isotherms at 77 K, BET surface areas of 1313 cm$^2$ g$^{-1}$ and 1173 cm$^2$ g$^{-1}$ were estimated for ZIF-76 and ZIF-76-mbIm, respectively. These values are consistent with those reported in the literature[23]. As expected, these crystalline ZIFs are also porous to a range of other small gases (Table 2, Supplementary Figures 23–28 and Supplementary Tables 2–4).

Remarkably, and representing a departure from the dense melt-quenched MOF glasses reported to date[16], these samples of $a_g$ZIF-76 and $a_g$ZIF-76-mbIm are permanently porous to incoming gases. While $a_g$ZIF-76 reversibly adsorbs in excess of 4 wt% $CO_2$ at 273 K (Supplementary Figure 29), isotherms measured on this glass were accompanied by significant hysteresis in their desorption branches, which persisted with long equilibration times. This indicates restricted diffusion of guest molecules due to constrictions in the pore network, which is consistent with very low uptake of $N_2$ and $H_2$ at 77 K. This is to be expected, given the partial collapse of the crystalline porous structure to the glass state. We therefore turned our attention to

the permanent porosity displayed by $a_g$ZIF-76-mbIm, which reversibly adsorbs $CO_2$ and $CH_4$ at 273 K and 293 K (Fig. 4a). In this case, the adsorption isotherms exhibited only very minor hysteresis, implying that diffusion limitations are largely absent for these gases. At 77 K, however, $N_2$ is prevented from diffusing into the pores, while $H_2$ is adsorbed but with significant hysteresis (Supplementary Figure 30).

The appreciable remnant network of accessible, interconnected pores in $a_g$ZIF-76-mbIm enables it to adsorb 7.0 wt% of $CO_2$ at 273 K and a pressure of 1 bar (Table 2). Micropore volumes and surface areas for ZIF-76-mbIm and $a_g$ZIF-76-mbIm were estimated by NLDFT fitting of their $CO_2$ adsorption isotherms (Supplementary Figures 31 and 32). We note that these are naturally lower than the values derived from $N_2$ isotherms measured at 77 K, which could only be obtained for ZIF-76-mbIm. The lower $CO_2$ uptake of $a_g$ZIF-76-mbIm compared to its crystalline precursor can be ascribed to the slight contraction of the overall pore volume of 0.12 cm$^3$ g$^{-1}$ upon vitrification. The enthalpy of adsorption at $Q_{st}$ for $CO_2$ indicates that $a_g$ZIF-76-mbIm binds $CO_2$ more strongly than ZIF-76-mbIm (Fig. 4b, Supplementary Figures 33 and 34). The modest $Q_{st}$ at zero coverage of the crystalline ZIF ($-27.2$ kJ mol$^{-1}$) arises from its largely non-polar pore environment, while the increase in $Q_{st}$ in the glass reveals a pore environment with enhanced interactions with the guest $CO_2$ molecules (vide infra). We ascribe the higher $Q_{st}$ for binding $CO_2$ of the glass to a highly contoured pore surface and constricted pore environment, which provides enhanced contacts with the guest. The relatively steep drop-off in $Q_{st}$ with increasing $CO_2$ loading of the glass indicates that the vitrification process creates regions within the pore network with a particularly high affinity for $CO_2$.

The kinetic profiles of $CO_2$ uptake were also measured for ZIF-76-mbIm and $a_g$ZIF-76-mbIm. Overadsorption of $CO_2$ initially leads to a fractional uptake above 1, before $CO_2$ release and equilibration. The $a_g$ZIF-76-mbIm takes significantly longer to attain equilibrium between the adsorbed and non-adsorbed gas molecules (Fig. 4c), which implies that diffusion is more constricted in the glass than its crystalline precursor, and the pore network is more tortuous. This is consistent with the reduced structural regularity, and hindered diffusion inherent in glass, with the remainder of its void space more closely resembling the parent material.

A further illustration of the pore structure is provided by the pore size distribution calculated from $CO_2$ adsorption isotherms at 273 K, which indicate that the pores contract slightly upon vitrification (Fig. 4d). Taken together, these observations show that glasses derived from ZIFs have potential in both kinetic and equilibrium-based gas separation processes. Glasses for gas separations may be optimized by matching the pore window size to the kinetic diameters of target gas pairs. Preliminary experiments indicate that this may be accomplished by varying linker ratios in these glasses. Vitrification of the crystalline frameworks is possible within a broad range of linker ratios to produce porous glasses with finely tuned textural characteristics. The adsorption of gases and the separation of gas mixtures using this suite of materials are the subjects of ongoing investigation.

We note that the network of channels in $a_g$ZIF-76-mbIm is stable for at least 3 months, a conclusion drawn from the reproducibility of adsorption isotherms measured on samples stored over this period of time. Further, the glasses can be handled in ambient laboratory atmospheres without any detrimental effects on their adsorption capacity. The superior textural characteristics of $a_g$ZIF-76-mbIm compared to $a_g$ZIF-76 can be ascribed to the presence of the methyl group in the former. As indicated by solid-state NMR, these groups anchor the glassy

**Table 2 Summary of the textural characteristics of a permanent porous glass derived from ZIF-76-mbIm**

|  | ZIF-76-mbIm | $a_g$ZIF-76-mbIm |
|---|---|---|
| Uptake of $CO_2$ (1 bar, 273 K) | 10.0 wt% | 7.0 wt% |
| Surface area from $N_2$ / 77 K data[a] | 1173 | d |
| Surface area from $CO_2$/273 K data[a] | 643 | 375 |
| Pore volume from $N_2$/77 K data[b] | 0.50 | d |
| Pore volume from $CO_2$/273 K data[b] | 0.17 | 0.12 |
| Isosteric heat of adsorption, $Q_{st}$ ($CO_2$)[c] | −26.3 | −29.3 |

[a]In cm$^2$ g$^{-1}$ using the BET model ($N_2$) or NLDFT fitting ($CO_2$)
[b]Volume accessible to adsorbate in cm$^3$ g$^{-1}$ at 1 bar using the density of liquid adsorbate ($N_2$) or NLDFT fitting
[c]In kJ mol$^{-1}$ at zero loading
[d]Not measurable due to diffusion limitations

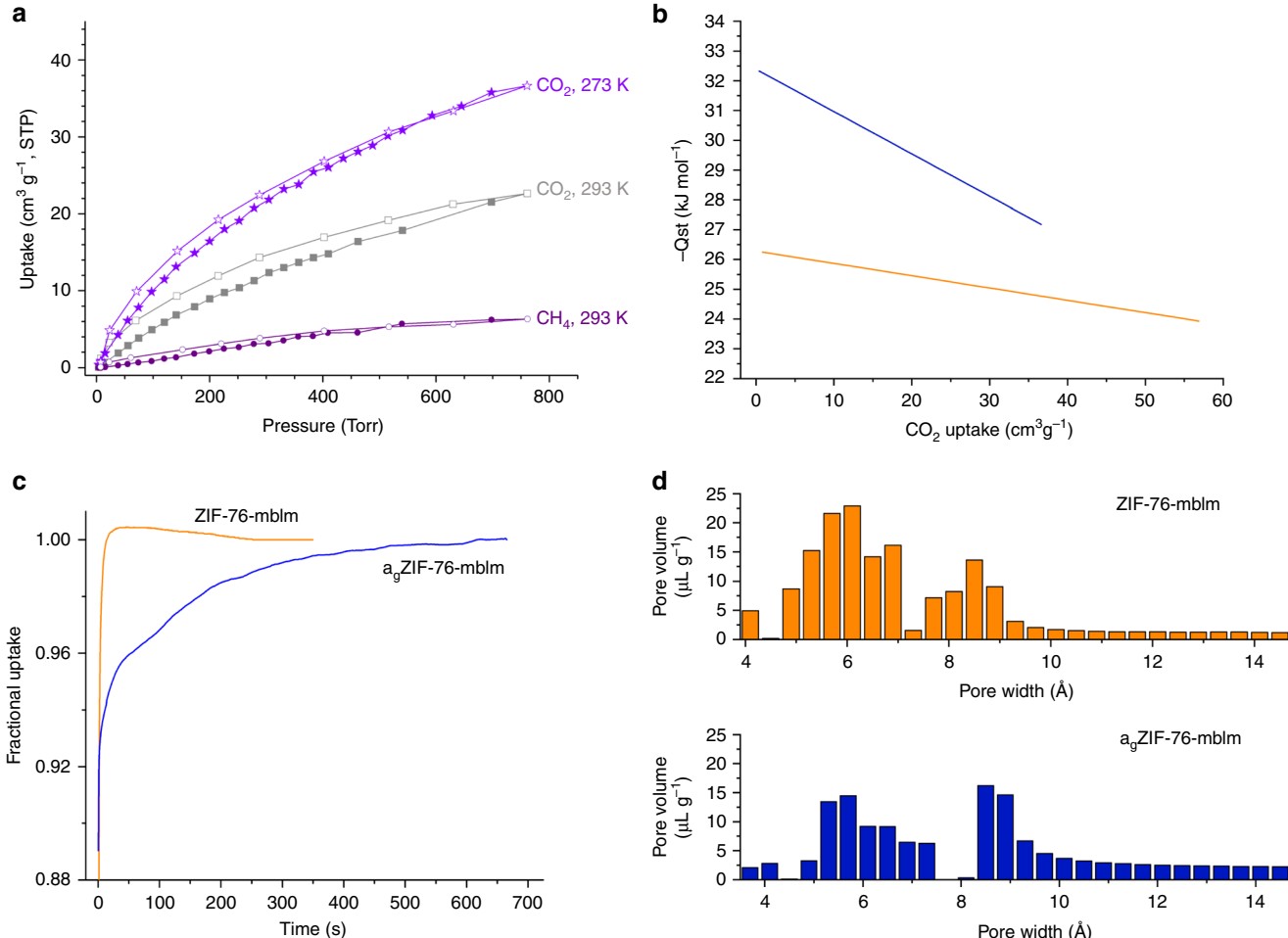

**Fig. 4** Permanent accessible porosity in a$_g$ZIF-76-mbIm and comparisons with ZIF-76-mbIm. **a** Adsorption isotherms of a$_g$ZIF-76-mbIm (filled symbols = adsorption, empty symbols = desorption). **b** Calculated isosteric heats of adsorption ($Q_{st}$) for $CO_2$ as a function of guest loading, orange ZIF-76-mbIm, blue a$_g$ZIF-76-mbIm. **c** Time-dependent $CO_2$ uptake profiles at 273 K at a pressure of 5 Torr. **d** Pore size distributions as determined by a NLDFT method from $CO_2$ adsorption isotherms at 273 K

network by non-covalent interactions with neighbouring ligands, which helps to maintain a relatively open and contiguous network of pores and channels.

## Discussion

This report describes permanent, accessible and reversible porosity inherent to glasses derived from metal–organic frameworks. Two precursor crystalline ZIFs were designed, and their high temperature melting monitored in situ by a range of combined diffraction experiments. Notably, these glasses are distinct from those reported by Yaghi et al.,[13] which are prepared via sol–gel methods, and in which $T_g$ disappears after solvent evaporation. This means that sol–gel glasses cannot be formed by heating to high temperatures. The discovery of accessible porosity in glasses derived from MOFs may serve as the foundation for a new class of porous hybrid inorganic–organic materials. We expect that developments in this field will be enabled by (i) the large number of known MOF or coordination polymer structures that can serve as potential glass precursors[32], (ii) the ability to combine the chemical diversity of MOFs with established techniques for handling and moulding glasses, (iii) the availability of several techniques for vitrifying crystalline frameworks[12,33] and (iv) the use of post-synthetic techniques that are employed in other glass families to increase available surface areas[34].

We envisage a plethora of potential applications will stem from porous MOF glasses, including membranes for chemical separations, catalysis, ion transport and conductivity[35]. These glasses should not however be placed into competition for the ultra-high surface areas heralded for crystalline MOFs, but seen in the light of ease of processing, mechanical stability and possible use in separations. Additional avenues for research may also arise from their comparison and contextualization with conventional glasses. In this light, MOF glasses may be geared towards applications in optics, where one of their principle advantages will lie in their softer nature and correspondingly lower processing temperatures. The combination of the stimuli responsivity of MOF chemistry[36] with the glass domain will also lead to new, smart applications and a new era of glass technology[37].

## Methods

**Synthesis**. ZIF-76 [Zn(Im)$_{1.62}$(5-ClbIm)$_{0.38}$] and ZIF-76-mbIm [Zn(Im)$_{1.33}$(5-mbIm)$_{0.67}$] were prepared via procedures developed by Peralta et al.[23]. Specifically, imidazole (0.12 g, 17.25 × 10$^{-4}$ mol) and 5-chlorobenzimidazole (0.13 g, 8.66 × 10$^{-4}$ mol) were mixed together in a solution of *N*,*N*-dimethylformamide (8.28 ml) and *N*,*N*-diethylformamide (5.73 ml). Zn(NO$_3$)$_2$.6H$_2$O (0.25 g, 8.59 × 10$^{-4}$ mol) was subsequently added, along with NaOH (0.52 ml, 2.5 mol dm$^{-3}$). The turbid solution was then heated to 90 °C for 5 days, and the microcrystalline powder collected by filtration. Occluded solvent was removed by heating under vacuum at 200 °C for 6 h[38]. For the mbIm equivalent, 5-methylbenzimidazole (0.115 g) was used in place of 5-chloroimidazole. Identical procedures were followed for samples

of $[Zn(Im)_{1.0}(5\text{-ClbIm})_{1.0}]$ and $[Zn(Im)_{0.93}(5\text{-mbIm})_{1.07}]$ except $NaOH_{(aq)}$ was omitted from the reaction solvent.

**Vitrification**. Bulk powder samples were placed into a ceramic crucible and then into a tube furnace, which was purged with argon, prior to heating at 10 °C min$^{-1}$ to the melting temperatures identified in Table 1. Upon reaching the set temperature, the furnace was turned off and the samples allowed to cool naturally (under argon) to room temperature.

**X-ray powder diffraction**. Data were collected with a Bruker-AXS D8 diffractometer using Cu Kα ($\lambda = 1.540598$ Å) radiation and a LynxEye position sensitive detector in Bragg–Brentano parafocusing geometry.

**Total scattering measurements**. X-ray data were collected at the I15-1 beamline at the Diamond Light Source, UK ($\lambda = 0.161669$ Å, 76.7 keV). A small amount of the sample was loaded into a borosilicate glass capillary of 1.17 mm (inner) diameter. Data on the sample, empty instrument and capillary were collected in the region of $\sim 0.4 < Q < \sim 26$ Å$^{-1}$. Background, multiple scattering, container scattering, Compton scattering and absorption corrections were performed using the GudrunX program[39,40]. Variable temperature measurements were performed using an identical set up, although the capillaries were sealed with araldite. Data were taken upon heating at 25 °C, 100 °C, 200 °C, 280 °C and then in 10 °C steps to 340 °C. Further data were collected in 20 °C intervals to 540 °C, before cooling and a final data set taken at room temperature. Data were corrected using equivalent data those taken from an empty capillary heated to identical temperatures.

**Combined small-angle–wide-angle X-ray scattering**. X-ray data were collected at the I22 beamline at the Diamond Light Source, UK ($\lambda = 0.9998$ Å, 12.401 keV). The SAXS detector was positioned at a distance of 9.23634 m from the sample as calibrated using a 100 nm period $Si_3N_4$ grating (Silson, UK), giving a usable $Q$ range of 0.0018–0.18 Å$^{-1}$. The WAXS detector was positioned at a distance of 0.16474 m from the sample as calibrated using a standard $CeO_2$ sample (NIST SRM 674b, Gaithersburg, MD, USA), giving a usable $Q$ range of 0.17–4.9 Å$^{-1}$. Samples were loaded into 1.5 mm diameter borosilicate capillaries under argon inside a glovebox and sealed to prevent the ingress of air. Samples were heated using a Linkam THMS600 capillary stage (Linkam Scientific, UK) from room temperature to 600 °C at 10 °C min$^{-1}$. Simultaneous SAXS/WAXS data were collected every 1 °C. Data were reduced to one dimensional (1D) using the DAWN package[41,42] and standard reduction pipelines[43]. Values for the power law behaviour of the samples were found using the power law model of SASView 4.1.1 (SasView version 4.1, 2017). Data were fitted over the range $0.003 \leq Q \leq 0.005$ Å$^{-1}$. Particle size distributions were calculated using the McSAS package[44,45], a minimal assumption Monte Carlo method for extracting size distributions from small-angle scattering data. Data were fitted over the range $0.002 \leq Q \leq 0.18$ Å$^{-1}$ with a sphere model.

**Nuclear magnetic resonance spectroscopy**. Solid-state NMR experiments were carried out on a 600 MHz Varian NMR system equipped with a 1.6 mm Varian HXY MAS probe. Larmor frequencies for [1]H and [13]C were 599.50 MHz and 150.74 MHz, respectively, and sample rotation frequency was 40 kHz. For 1D [1]H and [13]C MAS measurements, [1]H and [13]C 90° excitation pulses of 1.65 μs and 1.5 μs were used, respectively. In [1]H MAS NMR measurements 8 scans were co-added and repetition delay between scans was 3 s. In [13]C MAS NMR measurements number of scans was 5500 and repetition delay was 30 s. Frequency axes of [1]H and [13]C spectra were referenced to tetramethylsilane. [1]H-[13]C cross-polarization (CP) MAS NMR experiment employed Lee–Goldburg (LG) CP block with duration of 100 μs and high-power XiX heteronuclear decoupling during acquisition; number of scans was 660 and repetition delay between scans was 0.4 s. For two-dimensional [1]H–[1]H and [1]H–[13]C spin-diffusion measurements the numbers of scans were 16 and 4000, and numbers of increments in indirectly detected dimensions were 100 and 12, respectively. Repetition delay between scans was 0.5 s. During mixing periods of both measurements, the radio frequency driven dipolar recoupling (RFDR) scheme was used to enhance homonuclear dipolar coupling among protons. [1]H–[13]C experiment employed LG scheme during the CP block.

Solution [1]H NMR spectra of digested samples (in a mixture of DCl (35%)/$D_2O$ (0.1 ml) and DMSO-$d_6$ (0.5 ml)) of samples (about 6 mg) were recorded on a Bruker Avance III 500 MHz spectrometer at 293 K. Chemical shifts were referenced to the residual protio-solvent signals of DMSO-$d_6$. The spectra were processed with the MestreNova Suite.

**Differential scanning calorimetry and thermogravimetric analysis**. Characterizations of all the samples were conducted using a Netzsch STA 449 F1 instrument. The samples were placed in a platinum crucible situated on a sample holder of the DSC at room temperature. The samples were heated at 10 °C min$^{-1}$ to the target temperature. After cooling to room temperature at 10 °C min$^{-1}$, the second upscan was performed using the same procedure as for the first. To determine the $C_p$ of the samples, both the baseline (blank) and the reference sample (sapphire) were measured[46].

**Gas adsorption**. Isotherms were measured by a volumetric method using a Quantachrome Autosorb iQ2 instrument with ultra-high purity gases. Prior to analysis, the samples were degassed under a dynamic vacuum at $10^{-6}$ Torr for 10 h at 150 °C. Accurate sample masses were calculated using degassed samples after sample tubes were backfilled with nitrogen. Where possible, BET surface areas were calculated from $N_2$ adsorption isotherms at 77 K according to established procedures[47].

**Density measurements**. True densities were measured using a Micromeritics Accupyc 1340 gas pycnometer (1 cm$^3$ model). The typical mass used was 0.2 g, with the values quoted being the mean and standard deviation from a cycle of 10 measurements.

**Positron annihilation lifetime spectroscopy**. The $^{22}NaCl$, sealed in a thin Mylar envelope, was used as the source of positrons. The samples were packed to 2 mm thickness surrounding the positron source. The $o$-Ps lifetime measurements were taken under vacuum ($1 \times 10^{-5}$T orr) at 298 K using an EG&G Ortec spectrometer at a rate of $4.5 \times 10^6$ counts per sample. The lifetimes were converted to pore sizes by using the quantum-based formulation assuming a spherical pore geometry[48].

**Molecular simulations**. Grand canonical Monte Carlo simulations were performed as implemented in the RASPA simulation code[49] to compute single component gas adsorption isotherms of Ar (77 K), $CH_4$ (273 K), $CO_2$ (273 K), $H_2$ (77 K), $N_2$ (77 K) and $O_2$ (273 K) in ZIF-76 up to 1 bar (Supplementary Figure 35). The crystal structure of ZIF-76 $[Zn(Im)_{1.5}(clbIm)_{0.5}]$ was taken from the Cambridge Crystallographic Data Center (reference code = GITWEM)[50], and the disorder was removed manually to generate a suitable set of input coordinates for the calculations (Supplementary Data 1). Structural properties such as accessible pore volume, pore limiting diameter and the largest cavity diameter were calculated using Zeo++ software[51] and are listed in Supplementary Table 4. For pore volume calculations, the probe radius was set to zero.

## Data availability
The data that support the findings of this study are available from the corresponding authors on request.

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

## Acknowledgements

T.D.B. would like to thank the Royal Society for a University Research Fellowship, and for their support (UF150021). We acknowledge the provision of synchrotron access to Beamline I22 (exp. NT18236-1) at the Diamond Light Source, Rutherford Appleton Laboratory UK. We also thank Diamond Light Source for access to beamline I15-1 (EE171151). C.Z. acknowledges the financial support from the China Scholarship Council and the Elite Research Travel Scholarship from the Danish Ministry of Higher Education and Science. A.K. and G.M. acknowledge the support of the Slovenian Research Agency (research core funding No. P1-0021). C.M.D. is supported by the Australian Research Council (DE140101359) and a Veski Inspiring Women Fellowship. L.L. would like to thank the EPSRC for an allocated studentship. C.W.A. would like to thank the Royal Society for a PhD studentship (RG160498), and the Commonwealth Scientific and Industrial Research Organization (CSIRO) for additional support (C2017/3108). S.G.T., S.J.L. and O.T.Q. acknowledge the RSNZ Marsden Fund grant MAU1411. The authors gratefully acknowledge assistance from Dr. Shichun Li in data collection.

## Author contributions

T.D.B. designed the project and wrote the manuscript with S.G.T., and input from all authors. SAXS/WAXS experiments were performed by T.D.B., A.J.S., N.J.T. and G.J.S., with analysis by A.J.S. and G.J.S.. DSC measurements and sample preparation were carried out by C.Z., facilitated by Y.Y. PDF measurements carried out by T.D.B., D.A.K., C.A., L.L., C.W.A. and P.A.C. All gas adsorption measurements were carried out by S.G.T., O.T.Q. and S.J.L. Liquid NMR data were provided by A.Q., and solid-state NMR measurements carried out and analysed by G.M. and A.K. All simulations were carried out and analysed by I.E. PALS data were collected and analysed by C.M.D., A.W.T. and A.J.H.

## Additional information

**Competing interests:** The authors declare no competing interests.

