## [Peer Review File · Nature Communications]

REVIEWERS' COMMENTS:

Reviewer #1 (Remarks to the Author):

The authors have made modifications to the text but these are mainly clarifications. They are portraying porous MOF glasses as a family of solids where there is some relation between the parent MOF and the amorphous glass in that there are structural correlations under 0.6nm. Again, any zinc imidazolate will have zinc ions bridged by the anion so I am puzzled as to why this is noteworthy.

If the authors truly wish to make the case that this is an approach to a class of solids, they need to show more than one example. The rebuttal letter alludes to a different glass resulting when beginning with a different ZIF. For a Nature subjournal and a paper making a claim of a family of solids, I do not think it is too much to ask to present more than one example. In its present form, the manuscript could be accepted in a more specialized journal.

Reviewer #2 (Remarks to the Author):

The manuscript by Teller, Bennett, and coworkers presents a very interesting example of the preparation and characterization of MOF (ZIF) glasses with porosity. The authors have addressed the initial reservations I had that I felt should have been addressed prior to publication in [redacted]. This, in addition to the work they did to address other reviewer comments makes the current version of the manuscript absolutely publishable in Nature Communications.

I recommend publication of this manuscript in its current form.

REVIEWERS' COMMENTS:

Reviewer #1 (Remarks to the Author):

The authors have made modifications to the text but these are mainly clarifications. They are portraying porous MOF glasses as a family of solids where there is some relation between the parent MOF and the amorphous glass in that there are structural correlations under 0.6nm. Again, any zinc imidazolate will have zinc ions bridged by the anion so I am puzzled as to why this is noteworthy.

If the authors truly wish to make the case that this is an approach to a class of solids, they need to show more than one example. The rebuttal letter alludes to a different glass resulting when beginning with a different ZIF. For a Nature subjournal and a paper making a claim of a family of solids, I do not think it is too much to ask to present more than one example. In its present form, the manuscript could be accepted in a more specialized journal.

We are sorry that the referee is not in agreement with the three others approached previously. We present two examples of porous glasses in this paper – but we agree wholeheartedly with the referee that forming other porous glasses from MOFs would be an interesting approach for the future. We would nonetheless like to thank the referee for their time in reading, and commenting on the manuscript.

Reviewer #2 (Remarks to the Author):

The manuscript by Teller, Bennett, and coworkers presents a very interesting example of the preparation and characterization of MOF (ZIF) glasses with porosity. The authors have addressed the initial reservations I had that I felt should have been addressed prior to publication in [redacted]. This, in addition to the work they did to address other reviewer comments makes the current version of the manuscript absolutely publishable in Nature Communications. I recommend publication of this manuscript in its current form.

We thank the referee for their time and efforts in the reviewing process, and are pleased they feel able to support publication after revisions.